# Beneficial Effects of Two Hydrogen Sulfide (H_2_S)-Releasing Derivatives of Dexamethasone with Antioxidant Activity on Atopic Dermatitis in Mice

**DOI:** 10.3390/pharmaceutics15071907

**Published:** 2023-07-08

**Authors:** Silvia Abigail Coavoy-Sánchez, Anderson Romério Azevedo Cerqueira, Simone Aparecida Teixeira, Vincenzo Santagada, Giorgia Andreozzi, Angela Corvino, Antonia Scognamiglio, Rosa Sparaco, Giuseppe Caliendo, Beatrice Severino, Soraia Katia Pereira Costa, Luis Carlos Spolidorio, Marcelo Nicolás Muscará

**Affiliations:** 1Department of Pharmacology, Institute of Biomedical Sciences, University of São Paulo, São Paulo 05508-000, SP, Brazil; silviacoavoy@gmail.com (S.A.C.-S.);; 2Department of Pharmacy, School of Medicine, Università degli Studi di Napoli “Federico II”, 80131 Naples, Italy; 3Department of Physiology and Pathology, School of Dentistry, São Paulo State University, Araraquara 14801-903, SP, Brazil; 4Department of Physiology and Pharmacology, Cumming School of Medicine, University of Calgary, Calgary, AB T2N 4N1, Canada

**Keywords:** hydrogen sulfide, dexamethasone, atopic dermatitis, oxidative stress

## Abstract

Hydrogen sulfide (H_2_S) is particularly produced in the skin, where it participates in the regulation of inflammation, pruritus, cytoprotection, scarring, and angiogenesis. In this study, we compared the effects of dexamethasone (Dex) with two H_2_S-releasing Dex derivatives in a murine model of atopic dermatitis (AD) induced by topical application of 2,4-dinitrochlorobenzene (DNCB). After sensitization with DNCB, the animals were topically treated for five consecutive days with either the H_2_S-releasing compounds 4-hydroxy-thiobenzamide (TBZ) and 5-(p-hydroxyphenyl)-1,2-dithione-3-thione (ADT-OH), Dex, or the derivatives Dex-TBZ or Dex-ADT. Topical treatment with equimolar doses of either Dex, Dex-TBZ, or Dex-ADT resulted in similar reductions in dermatitis score, scratching behavior, edema, eosinophilia, splenomegaly, and histological changes. In contrast with Dex, the H_2_S-releasing derivatives prevented IL-4 elevation and oxidative modification of skin proteins. On an equimolar dose basis, Dex-TBZ, but not Dex-ADT, promoted the elevation of endogenous H_2_S production and GPx activity. Neither Dex-TBZ nor Dex-ADT decreased GR activity or caused hyperglycemia, as observed with Dex treatment. We conclude that the presence of H_2_S-releasing moieties in the Dex structure does not interfere with the anti-inflammatory effects of this corticosteroid and adds beneficial therapeutical actions to the parent compound.

## 1. Introduction

The chronic inflammatory skin disease atopic dermatitis (AD), also called eczema, is characterized by dry skin, pruritus, eczematous lesions, and impairment of the epidermal barrier, in addition to abnormal immune reactions, mainly due to IgE-mediated responses triggered by exogenous antigens [1]. The disease usually manifests during early infancy, but a substantial number of adults are also affected [2]; it is frequently associated with airborne allergies and asthma (atopic triad) [3].

AD has a complex and multifactorial etiology, as several genetic and environmental factors contribute to skin barrier dysfunction and immune dysregulation, which are considered central conditions for the pathogenesis of AD [4].

Topical treatment with corticosteroids, such as dexamethasone, is the first-line therapy for the management of many dermatologic conditions, including AD [5]. On the other hand, systemic therapy with conventional immunosuppressive drugs (including corticosteroids, azathioprine, ciclosporin, methotrexate, and mycophenolate mofetil), biologics, or Janus kinase inhibitors is the recommended therapeutical strategy when the disease is not properly controlled by topical treatments. However, systemic corticosteroid therapy is also the chosen alternative when large topical doses of these agents are needed because of the large surface area of skin affected by diseases or when long treatment periods are mandatory [6]. The beneficial effects of corticosteroids are related to their broad anti-inflammatory action, including decreased inflammatory cell infiltration into the tissues, which is attributed to the repression of pro-inflammatory genes that follows the activation of the intracellular glucocorticoid receptor [7]. However, it is important to emphasize that one of the big challenges of using corticosteroids is related to the adverse effects (such as skin atrophy, suppression of the hypothalamic–pituitary–adrenal axis, hyperglycemia, osteoporosis, etc.) that occur, especially, at high corticosteroid doses and/or prolonged use [8]. Based on these considerations, there is a clear need to develop better alternative treatments that improve the quality of life of AD patients.

Since the discovery of hydrogen sulfide (H_2_S) as the third gasotransmitter (after nitric oxide—NO and carbon monoxide—CO) [9], the relevance of this endogenous mediator as a signaling molecule in biological systems has been unanimously confirmed (for review, see Cao et al. [10]). Particularly in the skin, endogenous H_2_S participates in the regulation of important functions such as vasodilation, angiogenesis, cell proliferation, apoptosis, and inflammation (for review, see Coavoy-Sánchez et al. [11]). Previous studies have shown that H_2_S plays an important role in the pruritogenic and inflammatory response. For example, Rodrigues et al. [12] reported that the inhibition of endogenous H_2_S synthesis with β-cyano-L-alanine results in significant potentiation of the scratching behavior induced by the mast cell degranulator compound 48/80 in mice, in addition to increased neutrophil infiltration. On the other hand, the beneficial therapeutic actions of compounds capable of releasing H_2_S have been described in both histaminergic [12] and non-histaminergic pruritus in mice [13].

The chemical structures of these H_2_S donors have been exploited in the design of new therapeutic agents (for review, see Magli et al. [14]), among which it is worth mentioning H_2_S-releasing non-steroidal anti-inflammatory drugs (NSAIDs).

For example, the H_2_S-releasing derivative of naproxen, compound ATB-346, has been shown to be as effective as the parent drug in reducing inflammation, whilst producing significantly fewer gastrointestinal adverse effects [15,16].

Associations of H_2_S donors with corticosteroids have also been proposed, and studies in cells demonstrated that hybrid compounds of H_2_S donors and prednisone, betamethasone, or triamcinolone were more effective than the respective parent drug in inhibiting mast cell degranulation and in promoting membrane hyperpolarization of human bronchial smooth muscle cells. Furthermore, in a model of chronic allergic lung inflammation, treatment with a H_2_S-releasing prednisone compound was shown to be significantly more effective than prednisone to reduce peribronchiolar collagen density and smooth muscle thickness [17,18].

Considering the therapeutic potential of this type of association, we hypothesized that hybrid corticosteroid molecules, such as dexamethasone hybrids with H_2_S donors, may be promising therapeutic agents for controlling pruritus or other cutaneous and systemic inflammatory reactions that characterize AD, either enhancing the desired pharmacological effect or reducing the incidence of adverse effects of the original drug. In this way, the present study aims to compare the pharmacological effects of dexamethasone with two different H_2_S-releasing derivatives of dexamethasone (chemical structures shown in Figure 1) in a murine model of AD.

## 2. Material and Methods

### 2.1. Animals

All animal care and experimental procedures were approved by the local Ethics Committee for Animal Experimentation (CEUA-ICB/USP; n° 129/2016) and performed in accordance with the National Council for Control of Animal Experimentation (CONCEA) and the ARRIVE guidelines. Female Balb/c mice (20 ± 2 g; 6–8 weeks old) were purchased from the local USP animal breeding and care facilities and housed in polypropylene cages (in groups of five per cage) with free access to food (standard chow) and tap water, in a quiet room with controlled temperature (22 ± 2 °C) and a 12 h light–dark cycle.

### 2.2. Drugs

2,4-Dinitrochlorobenzene (DNCB), 4-hydroxy-thiobenzamide (TBZ), and dexamethasone (Dex) were obtained from Sigma-Aldrich Inc. (St. Louis, MO, USA); 5-(p-hydroxyphenyl)-1,2-dithione-3-thione (ADT-OH), dexamethasone-succinate-TBZ (Dex-TBZ), and dexamethasone-succinate-ADT (Dex-ADT) were synthesized in-house as previously described elsewhere [17,18].

### 2.3. Induction of Experimental AD and Drug Treatment

The mice were randomly divided into seven groups (*n* = 7 animals per group): (1) control (Sham), (2) vehicle-treated DNCB-induced AD, (3) Dex-treated DNCB-induced AD, (4) Dex-TBZ-treated DNCB-induced AD, (5) Dex-ADT-treated DNCB-induced AD, (6) TBZ-treated DNCB-induced AD, and (7) ADT-OH-treated DNCB-induced AD.

The mice were anesthetized with inhaled isoflurane (3% in O_2_, *v*/*v*; Matrix Medical, Orchard Park, NY, USA), and the dorsal region was shaved with an electric razor. Twenty-four hours later, sensitization was induced by the epicutaneous application of 200 μL of 0.5% DNCB in acetone/olive oil (3:1, *v*/*v*) on the shaved area, once daily over the first 3 days. On days 15, 17, 19, and 22, the mice were challenged by the application of 200 μL of 0.2% DNCB on the back skin and 20 μL of this solution on the dorsal side of the right ear. Sham control mice received the acetone/olive oil (3:1) vehicle alone. On days 19 to 23, following the sensitization procedure, the animals were topically treated once a day with a total dose of 250 nmol/mice (227 nmol on the dorsal skin + 23 nmol on the dorsal side of right ear) of either Dex, Dex-TBZ, or Dex-ADT (i.e., groups 3, 4, and 5, respectively), or with 1 μmol/mice of the H_2_S-releasing compounds TBZ and ADT-OH (groups 6 and 7, respectively; 909 nmol on the dorsal skin + 91 nmol on the dorsal side of right ear). Vehicle was topically applied both to the dorsal skin (200 μL) and the dorsal side of the right ear (20 μL) of groups 1 and 2.

On day 24, under anesthesia, the mice were weighed, and blood samples were collected by cardiac puncture and put into heparinized tubes. Plasma samples were obtained by centrifuging the blood samples at 1000× *g* (at 4 °C for 10 min) and were stored at −80 °C until analyzed. After euthanasia (obtained by isoflurane overdose), dorsal skin, ears, and spleens were removed for further analysis. The experimental protocol applied in the present study is schematically depicted in Figure 2.

### 2.4. Clinical Assessment of Dermatitis Severity

Dermatitis severity was evaluated in each animal immediately before each DNCB challenge, and on day 24, prior to animal euthanasia, when images of the skin lesions were captured with a digital camera. The scoring procedure consisted of analyzing the presence of each of the following signs: (i) erythema/hemorrhage, (ii) edema, (iii) excoriation/erosion, and (iv) scaling/dryness; these were scored as 0 (none), 1 (mild), 2 (moderate), or 3 (severe) and finally summed, according to Matsuda et al. [19].

### 2.5. Evaluation of Itching

Itching was assessed by the scratching behavior of the animals as analyzed before each DNCB challenge and on day 24, prior to animal euthanasia (always after the image capture). The animals were placed inside an acrylic cage in a room with acoustic isolation adapted for the use of a video camera. The scratching behavior was recorded for 30 min and the images were blindly analyzed by a trained investigator unaware of the treatments. The scratching behavior was quantified by counting the number of scratching bouts that occurred during the 30 min observation period. Each bout was considered as the event of continuous scratching of the animal back or ears by the hind paw, which ended when the hind paw was put into the mouth or rested on the bottom of the cage.

### 2.6. Ear Edema

The degree of ear edema was assessed by analyzing the ear water content. On day 24, after animal euthanasia, both ears were cut at the base and immediately weighed (wet weight). The tissues were then placed in an oven at 60 °C for 5 days and were then weighed again to obtain the corresponding dry weights. The percentage of water content of each ear was calculated as [(wet weight − dry weight)/wet weight] × 100%. Ear edema was expressed as the difference between the water content values of the right (treated) and the corresponding left (naïve) ear from each animal.

### 2.7. Histopathological Analysis

The excised dorsal skin tissue samples were fixed in 4% formalin (Sigma-Aldrich, St. Louis, MO, USA) for 24 h and subsequently embedded in paraffin. The tissues were sliced into 5 μm thick sections and mounted onto glass slides for further staining with either hematoxylin and eosin (H&E) or toluidine blue (TB), as previously described [20]. H&E-stained slides were used for morphometric analysis of epidermal thickness and leukocyte count. Toluidine blue was used for the measurement of mast cell infiltration. The stereometric analysis was performed by a single blinded trained examiner, using a point-counting technique according to the methods employed in previous studies [21]. The analysis was carried out using cuts of the vehicle group as a reference of normality. The results were expressed as a percentage of the total area analyzed. Images of the sections were captured using a digital camera coupled to a transmitted light microscope (Leica Microsystems GmbH, Wetzlar, Germany) at 200× magnification, and analyzed using an image analysis program Fiji ImageJ (National Institutes of Health, Bethesda, MD, USA).

### 2.8. Circulating Leukocyte Counting

The number of total leukocytes was determined by counting total white blood cells in a Neubauer chamber after a 1:40 dilution of the anticoagulated blood with Turk’s solution (0.2% violet crystal in 30% aqueous acetic acid solution). The percentage of blood eosinophils was determined by optical microscopy (based on the differential leukocyte nucleus morphology) after May–Grünwald–Giemsa staining of blood smears.

### 2.9. Spleen Weight Measurement

The spleens were carefully excised and weighed, and each value was normalized by the respective mouse body weight.

### 2.10. ELISA Measurements

Plasma IgE concentrations were quantified using a commercial ELISA kit (ELISA MAX™ Deluxe Set kit, BioLegend, San Diego, CA, USA) following the manufacturer’s instructions.

The skin samples were homogenized in TRIS-HCl buffer (50 mM, pH 7.4) containing 1 mM phenylmethylsulfonyl fluoride (PMSF) and 0.1% protease inhibitor mixture (P8340, Sigma, St. Louis, MO, USA), at 4 °C using a micro grinder. After centrifugation (10,000× *g* for 10 min at 4 °C), IL-4 and eotaxin-1 concentrations were quantified in the supernatants with the aid of the respective ELISA (ELISA MAX™ Set (BioLegend, San Diego, CA, USA)), according to the manufacturer’s instructions. The resulting absorbance values (at 450 nm with correction at 570 nm) were read using a microplate reader (SpectraMax Plus 384, Molecular Devices, Sunnyvale, CA, USA), and each cytokine concentration was calculated by extrapolation of the absorbance values from the standard curve prepared using the respective recombinant cytokine.

### 2.11. Measurement of Endogenous H_2_S Production

The rate of H_2_S production by the skin samples ex vivo was measured according to the previously reported lead acetate/lead sulfide method [12].

Briefly, dorsal skin tissues were excised and homogenized in phosphate buffer (100 mM, pH 7.4) containing 1 mM PMSF and 0.1% protease inhibitor mixture at 4 °C using a micro grinder. After centrifugation (10,000× *g*; 10 min; 4 °C), each supernatant (containing 200 μg protein) was plated in a 96-well plate and supplemented with 10 mM L-cysteine and 2 mM pyridoxal-5-phosphate (substrate and co-factor, respectively). A filter paper previously embedded with 100 mM lead acetate (and allowed to dry) was used to cover the plate which was further incubated for 3 h at 37 °C. At the end of this period, the dark spots developed on the filter paper (due to the formation of a dark brown lead sulfide—PbS precipitate) were densitometrically analyzed using the software ImageLabTM (Bio-Rad, Hercules, CA, USA). A standard curve prepared with NaHS in phosphate buffer (concentration range: 7.8–500 μM) was submitted to the same procedures and used to calculate the H_2_S production by each skin homogenate sample.

### 2.12. Antioxidants Enzymes

Fragments of the collected dorsal skin samples were homogenized in phosphate buffer (50 mM, pH 7.0) containing 100 mM ethylenediaminetetraacetic acid (EDTA) and centrifuged (10,000× *g;* 10 min; 4 °C). The supernatants were collected and used for enzyme activity assays. Protein concentration was measured using the Bradford assay [22].

Total superoxide dismutase (SOD) activity was estimated by the rate of inhibition of XTT (3′-{1-[(phenylamino)-carbonyl]-3,4-tetrazolium}bis(4-methoxy-6-nitro)benzenesulfonic acid hydrate) reduction at 470 nm, as previously described by Ukeda et al. [23], with modifications. Results are expressed as units (U) of SOD/mg of protein; one unit of SOD is defined as the amount of enzyme capable of dismutating 1 μmol O_2_^−^/min.

Catalase activity was measured by the decrease in the concentration of H_2_O_2_ at 460 nm, as previously described by Fossati et al. [24]. Results were expressed as units of catalase/mg of protein; one catalase unit is defined as the amount of H_2_O_2_ (μmol) degraded per min.

Glutathione peroxidase (GPx) activity was determined by measuring the rate of formation of oxidized glutathione (GSSG) from reduced glutathione (GSH) in the presence of tert-butyl hydroperoxide, which was detected by the decrease in reduced nicotinamide adenine dinucleotide phosphate (NADPH) at 340 nm, as previously described [25]. Results were expressed as μmol GSH/min/mg protein.

Glutathione reductase (GR) activity was measured by monitoring the decrease in absorbance at 340 nm by the oxidation of NADPH, which acts as a cofactor in the reduction of GSSG to GSH by GR, as previously described [26]; one unit of GR catalyzes the oxidation of 1 μmol NADPH/min. Results were expressed as μmol NADPH/min/mg of protein.

Glutathione S-transferase (GST) activity was determined by measuring the rate of conjugation of GSH with 1,2-dichloro-4-nitrobenzene, detected by the increase in absorbance at 340 nm, as previously described [27]. Results were expressed as μmol GSH/min/mg protein.

### 2.13. Analysis of Protein 3-Nitrotyrosine Residues and Carbonyl Groups

The presence of proteins containing 3-nitrotyrosine (NT) residues or carbonyl groups in the collected skin samples was assessed by slot blotting. NT-containing proteins were analyzed using a mouse monoclonal anti-nitrotyrosine antibody (diluted 1:2000; Merck Millipore Co., Darmstadt, Germany), as previously described [28].

The presence of carbonylated proteins was analyzed by the method previously described by Robinson et al. [29]. Briefly, after sample protein blotting, the membranes were treated with 2,4-dinitrophenylhydrazine (DNPH; 0.1 mg/mL in 2 N HCl, 5 min) in order to obtain the protein hydrazone derivatives. After several washing steps, the membranes were incubated overnight at 18 °C with a primary rabbit anti-dinitrophenol (DNP) antibody (Abcam, Cambridge, UK; diluted 1:25,000 in blocking buffer).

Immunoreactivity of the blots was detected using secondary HRP-conjugated antibodies, and after incubation with a chemiluminescence reagent, chemiluminescence intensities were quantified by densitometric analysis (ImageLabTM, Bio-Rad Laboratories, Inc., Hercules, CA, USA). Results were normalized by the intensity values obtained after staining the slots with a Ponceau S dye solution.

### 2.14. Clinical Biochemistry

Plasma glucose, total bilirubin, aspartate aminotransferase (AST), alanine aminotransferase (ALT), and gamma glutamyl transpeptidase (GGT) were quantified by colorimetric assays using the respective commercial kits (from Labtest, Lagoa Santa, MG, Brazil): Glucose Liquiform (Ref. 133), Bilirubin (Ref. 31), AST/GOT Liquiform (Ref. 109), ALT/GPT Liquiform (Ref. 109), and Gama GT Liquiform (Ref. 105).

### 2.15. Statistical Analysis

Data are expressed as mean ± standard error of the mean (S.E.M.). With the aid of the software GraphPad Prism (v.9.3.0), time-course data were analyzed by two-way ANOVA followed by Bonferroni’s test. Differences among the means of the different experimental groups were analyzed by one-way ANOVA followed by Dunnett’s test for multiple comparisons. Values of *p* < 0.05 were considered statistically significant.

## 3. Results

### 3.1. Effects of Treatments on Clinical AD Signs

As shown in Figure 3, repeated application of 200 μL of 0.2% DNCB to the dorsal skin of mice previously sensitized with 0.5% DNCB promoted a significant and time-dependent increase in the characteristic clinical signs of AD, such as skin lesions (Figure 3A) and pruritus (Figure 3B) from day 19, as well as a significant increase in ear edema on day 24 (Figure 3C), compared with the Sham group.

Topical treatment for five consecutive days (from day 19 to day 23) with both Dex-TBZ and Dex-ADT hybrid donors (250 nmol/mice) exhibited effects comparable to the equimolar dose of Dex in reducing dermatitis score, scratching behavior, and ear edema induced by repeated application of DNCB to the skin of mice. On the other hand, administration of the H_2_S-releasing moieties, TBZ or ADT-OH donors (1 μmol/mice), showed no change in DNCB-induced dermatitis, pruritus, and ear edema.

### 3.2. Histopathological Analysis

Histological evaluation of dorsal cutaneous tissue samples after H&E staining showed that the topical application of DNCB to the dorsal skin led to epidermal hyperplasia, spongiosis, hyper-parakeratosis, and dense dermal infiltration of mononuclear and polymorphonuclear leukocytes (Figure 4A). Topical treatments with either Dex, Dex-TBZ, or Dex-ADT significantly decreased the epidermal thickness (Figure 4C) and substantially reduced the inflammatory infiltrate (Figure 4D) when compared to the untreated AD group, although Dex-TBZ and Dex-ADT effects were significantly higher than those of Dex on epidermal thickness; treatments with either TBZ or ADT-OH reduced the morphological alterations of the epidermis and dermis in a less accentuated way. Staining of skin sections with toluidine blue (Figure 4B) revealed increased mast cell infiltration into the dermis of untreated AD mice, in comparison with the control Sham group, whereas topical application of either Dex, Dex-TBZ, or Dex-ADT reduced this infiltration (Figure 4E); treatment with either TBZ or ADT-OH failed to significantly reduce mast cell infiltration.

### 3.3. Effects of Treatments on DNCB-Induced Eosinophilia and Splenomegaly

As shown in Figure 5, the application of DNCB significantly increased the number of total white blood cells (WBC—Figure 5A), blood eosinophils (Figure 5B), and spleen weight (Figure 5C) in comparison with the Sham group.

Topical treatment with either Dex, Dex-TBZ, or Dex-ADT significantly decreased the numbers of WBC and eosinophils, as well as the splenomegaly induced by DNCB, with values below those observed in animals without AD. Furthermore, topical treatment with TBZ or ADT-OH decreased DNCB-induced eosinophilia, and ADT-OH significantly reduced splenomegaly compared to the experimental AD group.

### 3.4. Plasma IgE Concentration and Skin IL-4 and Eotaxin-1 Contents

As shown in Figure 6, experimental AD induction led to a significant increase in total serum IgE concentrations (panel A), as well as skin IL-4 and eotaxin-1 contents (panels B and C, respectively) when compared with the control Sham group.

None of the treatments affected plasma IgE or skin eotaxin-1. Topical treatment with Dex or Dex-TBZ did not result in significant changes in skin IL-4 in relation to the DNCB group; however, skins obtained from animals treated with Dex-TBZ contained significantly lower amounts of this cytokine when compared with the Dex group. Treatment with Dex-ADT or the H_2_S-releasing compounds TBZ and ADT-OH significantly reduced skin IL-4 contents when compared with the untreated DNCB group (Figure 6B).

### 3.5. Effects of Treatments on Skin H_2_S Production

As shown in Figure 7, the induction of experimental AD resulted in decreased ex vivo skin H_2_S production. This decrease was reversed in the Dex-TBZ-treated group, which also showed significantly higher H_2_S production than the Dex- or Dex-ADT-treated groups.

### 3.6. Antioxidant and Oxidative Stress Markers

As shown in Figure 8, SOD (Figure 8A), catalase (Figure 8B), and GST (Figure 8E) enzyme activities were significantly reduced in the skin of animals with experimental AD when compared with the control Sham group, and none of the treatments affected these activities. However, as shown in Figure 8C, treatment with either Dex, Dex-TBZ, or ADT-OH led to increased GPx activity, although Dex-TBZ effects were significantly higher than those of Dex. Skin GR activity was not affected by the presence of AD; however, it was found significantly reduced in the Dex-treated animals (Figure 8D).

The presence of AD resulted in oxidative damage to skin proteins, as assessed by protein carbonylation and 3-nitrotyrosine contents (Figure 8F), and these responses were significantly prevented by either Dex-TBZ or Dex-ADT, but not by the other treatments.

### 3.7. Clinical Biochemistry

Mice with AD did not show different plasma glucose concentrations than those found in the control Sham group. Treatment with Dex, but not Dex-TBZ or Dex-ADT, resulted in significant hyperglycemia (Figure 9A).

With regard to circulating liver enzymes (Figure 9C), GGT remained unaltered among all the experimental groups. On the other hand, circulating ALT and AST activities were significantly elevated in the untreated animals with experimental AD. While treatment with the H_2_S donors TBZ and ADT-OH significantly prevented the ALT elevation, treatment with either Dex, Dex-TBZ, or Dex-ADT significantly potentiated the AD effects on this enzyme. Regarding AST, none of the treatments affected the activity of this enzyme in comparison with the untreated DNCB group.

## 4. Discussion

Topical and systemic corticosteroids are commonly used as anti-inflammatory therapy for AD [5,6]; however, long-term treatment with these compounds is limited due to several side effects, thus making it necessary to develop alternative strategies.

Many studies provide evidence of the pathophysiological role of H_2_S in the skin [11]. The expression of the H_2_S-producing enzymes, cystathionine-γ-lyase (CSE), cystathionine-β-synthase (CBS), and 3-mercaptopyruvate sulfurtransferase (3MST), has been reported in skin homogenates, and in some cases, associations between endogenous H_2_S production and the severity of skin disease, or atopic conditions, have been proposed. For example, Alshorafa et al. [30] found that psoriasis, a chronic inflammatory skin disease, is associated with low serum concentrations of H_2_S. Previous results from our laboratory evidence that inhibition of endogenous H_2_S synthesis potentiates itching secondary to mast cell degranulation induced by compound 48/80 in mice [12]. Wu et al. [31] reported that plasma H_2_S concentrations in asthmatic patients are lower than those measured in healthy subjects. However, a recent study by Moniaga et al. [32] observed that patients with AD have elevated concentrations of serum H_2_S and increased expression of CSE, CBS, and 3MST. These findings may support the concept that the benefits of topical application of exogenous H_2_S (such as those frequently obtained by patients during balneotherapy in natural waters containing high sulfide concentrations) are not always meant to compensate for the diminished endogenous production of H_2_S, but to make the most of its effects.

Several lines of research suggest the use of exogenous H_2_S (as H_2_S donors) for the treatment of a variety of inflammatory conditions, mainly dermatological diseases, based on the observed anti-inflammatory, antipruritic, pro-healing, pro-angiogenic, and pro-resolving effects [11]. However, exogenous H_2_S can show opposite effects depending on the administered dose, as is the case for pruritus: while low H_2_S doses are anti-pruritic [12,13], at high doses, H_2_S induces itching [33].

In this way, the chemical characteristics of the H_2_S donors are of central relevance, mainly due to the rate of H_2_S release. Regarding the H_2_S-releasing donors coupled to Dex, TBZ and ADT-OH, both release H_2_S after hydrolysis, and in comparison with inorganic sulfide salts (spontaneous releasers), they supply a slow and sustained H_2_S release. As shown by the chemical structures (Figure 1), the TBZ moiety has a single sulfur atom, meaning that 1 mol of H_2_S is released per mol of compound after its hydrolysis. On the other hand, ADT-OH contains three sulfur atoms and more than 1 mol of H_2_S per mol of compound can be sequentially released [34], although in the present study, both Dex derivatives showed comparable effects.

H_2_S donors have also been explored as moieties coupled to the structure of traditional NSAIDs (thus giving rise to new hybrid molecules), capable of reducing inflammation as effectively as the parent drugs, although devoid of the harmful effects on the gastrointestinal tract [15,35,36]. Hybrid H_2_S-releasing corticosteroids were also synthesized and proved to be more effective than the parent drugs in experimental asthma [17,18]. Thus, in the present work, we decided to comparatively evaluate the effects of dexamethasone in relation to those of the hybrid H_2_S-donor derivatives Dex-TBZ and Dex-ADT, as well as the respective H_2_S-releasing moieties alone, TBZ and ADT-OH, when applied topically in a widely accepted experimental model of AD in mice, in addition to at least some of the pharmacological mechanisms involved and possible adverse effects.

The mouse model of AD in the present study was characterized by clinical and histological features similar to those found in human AD, including pruritus, eczematous skin lesions (erythema, edema, dryness), epidermal hyperplasia, spongiosis, parakeratosis, and chronic inflammatory infiltrate (mainly lymphocytes, mast cells, and eosinophils) [2,37]. Skin lesion scores, pruritus, and ear edema were initially evaluated in mice with experimental AD. Topical treatment with 250 nmol/animal of Dex, Dex-TBZ, or Dex-ADT resulted in significant reductions in pruritus, dermatitis score, and ear edema induced by repeated applications of DNCB. The administration of the H_2_S-releasing moieties, TBZ or ADT-OH, at 1 μmol/animal did not result in any significant reduction in these clinical signs. These results evidence that the derivatization of Dex with either TBZ or ADT-OH does not impair the beneficial effects of the parent compound.

Furthermore, our results show that Dex, Dex-TBZ, and Dex-ADT ameliorated the histological changes induced by AD, such as epidermal thickness and increased leukocyte and mast cell infiltration in the skin, although Dex-TBZ and Dex-ADT were more effective in comparison with Dex. Treatment with TBZ or ADT-OH donors alone also reduced epidermal thickness and leukocyte infiltration. In fact, these results are in agreement with previously published studies that demonstrate that H_2_S can reduce inflammatory cell infiltrations [17,38], and that these actions can be partly due to the inhibition of leukocyte adherence to the vascular endothelial cells via activation of ATP-sensitive K^+^ (K_ATP_) channels [38].

Blood and tissue eosinophilia are common features of human AD that appear to directly correlate with disease activity [39]. In this work, the increase in the number of blood eosinophils secondary to AD in mice was completely inhibited by treatment with Dex, Dex-TBZ, or Dex-ADT, with values even lower than those observed in the control sham animals. Remarkably, these results are consistent with the known immunosuppressive activity of dexamethasone and other corticosteroids, which dramatically reduce the number of lymphocytes, monocytes, and eosinophils in the blood by direct induction of apoptosis [40]. Interestingly, topical treatment with the H_2_S donors TBZ or ADT-OH also led to significantly decreased DNCB-induced blood eosinophilia. Inhibitory effects of eosinophilia by H_2_S donors have been previously reported; in fact, animals with allergic asthma (induced by ovalbumin) treated with NaHS present with a decreased number of eosinophils in the bronchoalveolar lavages [41,42,43], as well as lower eosinophil influx into the lung parenchyma [44].

IgE is released in response to various allergens, and thus, IgE is an immunological hallmark of AD. Th2 cells produce cytokines IL-4 and IL-13, which are involved in the excessive production of IgE. Subsequently, IgE binds to allergens and mast cells to induce their degranulation and the release of inflammatory mediators [45,46].

In the present study, the increase in plasma IgE in mice with experimental AD was not reduced by the topical treatment with either Dex, Dex-TBZ, Dex-ADT, TBZ, or ADT-OH, which may be a consequence of the short-term treatment. Interestingly, the elevated levels of IL-4 in the dorsal skin lesions of mice with AD were significantly reduced by treatment with Dex-ADT, but not with Dex or Dex-TBZ, although the animals treated with the latter showed significantly lower concentrations than those measured in the Dex-treated group.

It is well known that corticosteroids suppress the synthesis of Th2-derived cytokines (such as IL-4 and IL-13) necessary for the production of IgE, thus contributing to their effectiveness in controlling allergic diseases. Interestingly, corticosteroids may also alter the Th1/Th2 balance in favor of the Th2 cell dominance by, probably, suppressing interferon (IFN)-γ, which normally inhibits Th2 differentiation [47,48]. In this instance, the hybrid molecules seem to avoid the harmful effect of dexamethasone. Furthermore, treatment with the H_2_S donors TBZ and ADT-OH alone also resulted in a significant reduction in IL-4 concentrations. It is noteworthy that these results are consistent with the reported ability of H_2_S to attenuate Th2 cytokine production [43,49].

We also analyzed the effects of H_2_S donors on mouse skin H_2_S production. Interestingly, treatment of the animals with Dex-TBZ, but not Dex-ADT or Dex, resulted in the reversal of the decreased in vitro H_2_S production by the skin of untreated mice with AD. This effect could be related to an increase in the expression of H_2_S-generating enzymes by exogenous H_2_S. In fact, Wu et al. [50], using a murine model of accelerated aging induced by D-galactose, observed that treatment of the animals with NaHS led to increased expression of CSE and CBS in the heart, liver, and kidney.

Oxidative stress plays an important pathogenetic role in human AD [51,52,53,54]. In order to evaluate the involvement of oxidative stress in our murine model of AD, the activity of antioxidant enzymes and the expression of oxidative stress markers (such as 3-nitrotyrosine and protein carbonylation) were determined in the mouse skin samples. In the present study, we found that the activities of SOD, catalase, and GST enzymes are significantly reduced in the skin of mice with experimental AD, in addition to significantly increased protein nitration and carbonylation.

It is well known that H_2_S has antioxidant and cytoprotective activities in physiological systems exposed to reactive oxygen and nitrogen species (ROS and RNS, respectively). H_2_S exerts its antioxidant effects via several mechanisms, including direct quenching of ROS, modulation of GSH and thioredoxin expression and activity, or increased synthesis of antioxidant proteins secondary to activation of the transcription factor Nrf2 [55,56,57]. In addition, Nrf2 has a role in the epidermal barrier function to protect against oxidative damage [58].

In the present study, treatment with Dex, Dex-TBZ, and ADT-OH significantly increased GPx activity in the skin of animals with experimental AD, with the effects of Dex-TBZ being significantly more intense than those of Dex. In addition, despite treatment with Dex having led to diminished GR activity, this effect was prevented in the groups of animals treated with Dex-TBZ or Dex-ADT. As a whole, treatment with the H_2_S-releasing Dex derivatives, but not the parent compound, results in the improvement of the endogenous antioxidant systems responsible for H_2_O_2_ elimination, by both increasing GPx-mediated GSH peroxidation and maintaining adequate GSH concentrations for the latter reaction via increased GR.

Furthermore, treatment with Dex-TBZ or Dex-ADT, but not Dex, was also able to prevent protein nitration and carbonylation in skin tissues. In this context, these results show that the addition of the H_2_S-releasing moieties, TBZ or ADT-OH, to Dex stimulates antioxidant defenses, thus decreasing ROS- and RNS-mediated oxidative modifications in the skin and providing greater protection against oxidative stress-induced damage. In agreement with our results, some studies have previously shown that H_2_S donors have potent antioxidant effects, as assessed in models of dermatitis and other atopic diseases. For example, Wu et al. [59] observed that sulforaphane (a sulfur-containing compound) alleviated the symptoms of DNCB-induced AD in mice, through activation of the Nrf2 factor and suppression of JAK1/STAT3 signaling. In a mouse model of allergic asthma, Benetti et al. [42] demonstrated that treatment with NaHS is able to prevent pulmonary allergic inflammation by increasing the expression of SOD, GR, and GPx enzymes.

Although splenomegaly is not a clinical sign associated with AD in humans, this alteration is present in the herein-employed murine model of AD [60,61,62]. In this study, we show that splenomegaly secondary to experimental AD is completely inhibited by topical treatment with different doses of either Dex, Dex-TBZ, or Dex-ADT. In fact, it has already been reported that in this animal model, corticosteroids exert inhibitory effects on lymphoid organs, such as the thymus and spleen, probably by increasing the rate of apoptosis of thymocytes and splenocytes [63]. Interestingly, treatment with ADT-OH alone also reduced DNCB-induced splenomegaly, evidencing the immunosuppressive effects of this H_2_S donor.

Other clinical unwanted side effects of corticosteroids include increased blood glucose and liver enzymes (ALT and AST). Corticosteroids can interfere with glucose metabolism, as they inhibit peripheral glucose uptake in muscle and adipose tissues, thus antagonizing insulin response. In addition, they increase hepatic gluconeogenesis, which also contributes to hyperglycemia, resulting in additional risk for diabetic patients [64]. In the present study, a significantly increased glycemia was observed in the Dex-treated animals, but remained unaltered in those animals treated with the H_2_S-releasing derivatives, Dex-TBZ and Dex-ADT, or even with the H_2_S donors TBZ or ADT-OH alone. In fact, some studies demonstrate that H_2_S can directly contribute to the homeostatic maintenance of blood glucose concentrations [65]. The regulation of glucose metabolism promoted by H_2_S is complex and occurs at several levels, including the production/release of insulin by pancreatic β cells and the response of target organs (such as liver, skeletal muscle, and adipose tissue) to insulin [66,67,68,69,70].

We also observed increased circulating liver enzyme (ALT and AST) activities in animals with experimental AD, which was similarly potentiated by Dex and its H_2_S-releasing derivatives, indicating corticosteroid-induced hepatic effects. Interestingly, this potentiation did not occur in the animals treated with the H_2_S donors TBZ or ADT-OH, but rather, increased serum AST in AD mice was normalized by these treatments. In fact, a study conducted by Wei et al. [71] demonstrated that circulating H_2_S concentrations are negatively correlated with liver enzyme ALT and AST activities, suggesting a protective role for H_2_S on liver function.

## 5. Conclusions

The presence of H_2_S-releasing moieties TBZ or ADT-OH in the dexamethasone molecule does not interfere with the beneficial effects of this corticosteroid but rather adds beneficial actions over treatment with the parent molecule, such as the increase in antioxidant enzymes (and the consequent avoidance of oxidative protein modifications that occur in AD) and the prevention of hyperglycemia induced by the parent corticosteroid dexamethasone. In view of the wide array of inflammatory conditions that routinely make use of corticosteroids, our data not only evidence the therapeutic potential of this new class of anti-inflammatory agents but also stimulate and support the performance of future clinical trials.

## Figures and Tables

**Figure 1 pharmaceutics-15-01907-f001:**
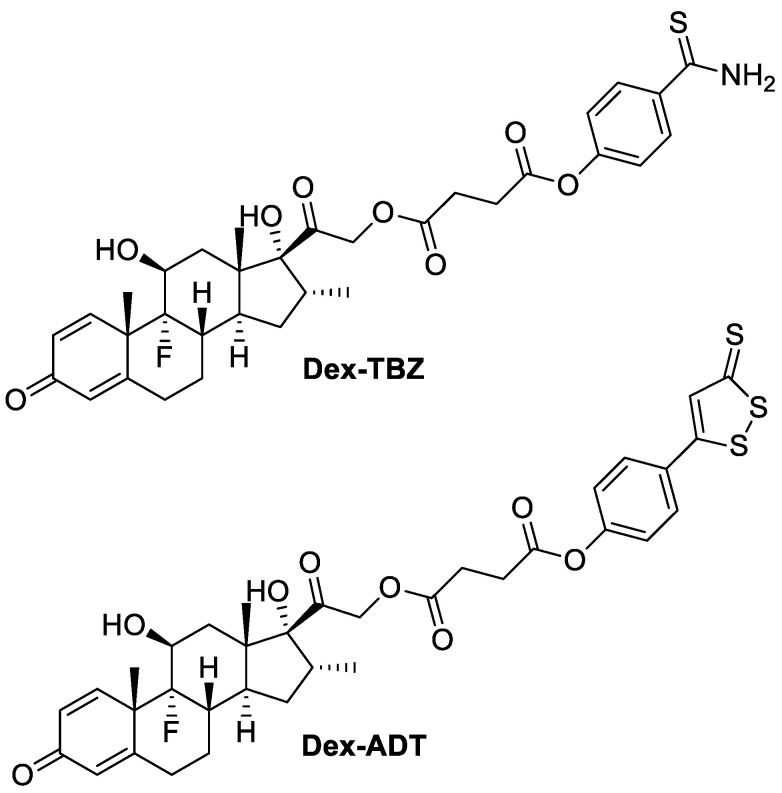
Chemical structures of the H_2_S-releasing dexamethasone derivatives.

**Figure 2 pharmaceutics-15-01907-f002:**
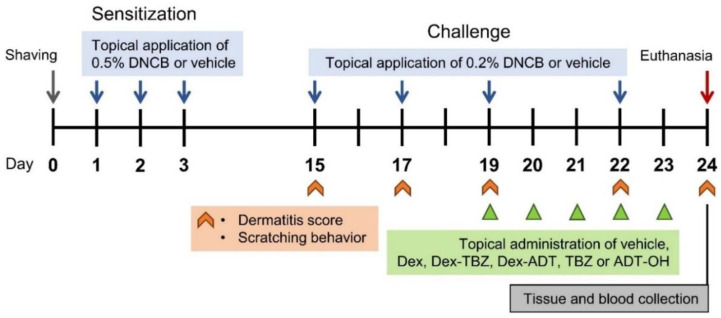
Experimental scheme for experimental AD induction and drug treatments.

**Figure 3 pharmaceutics-15-01907-f003:**
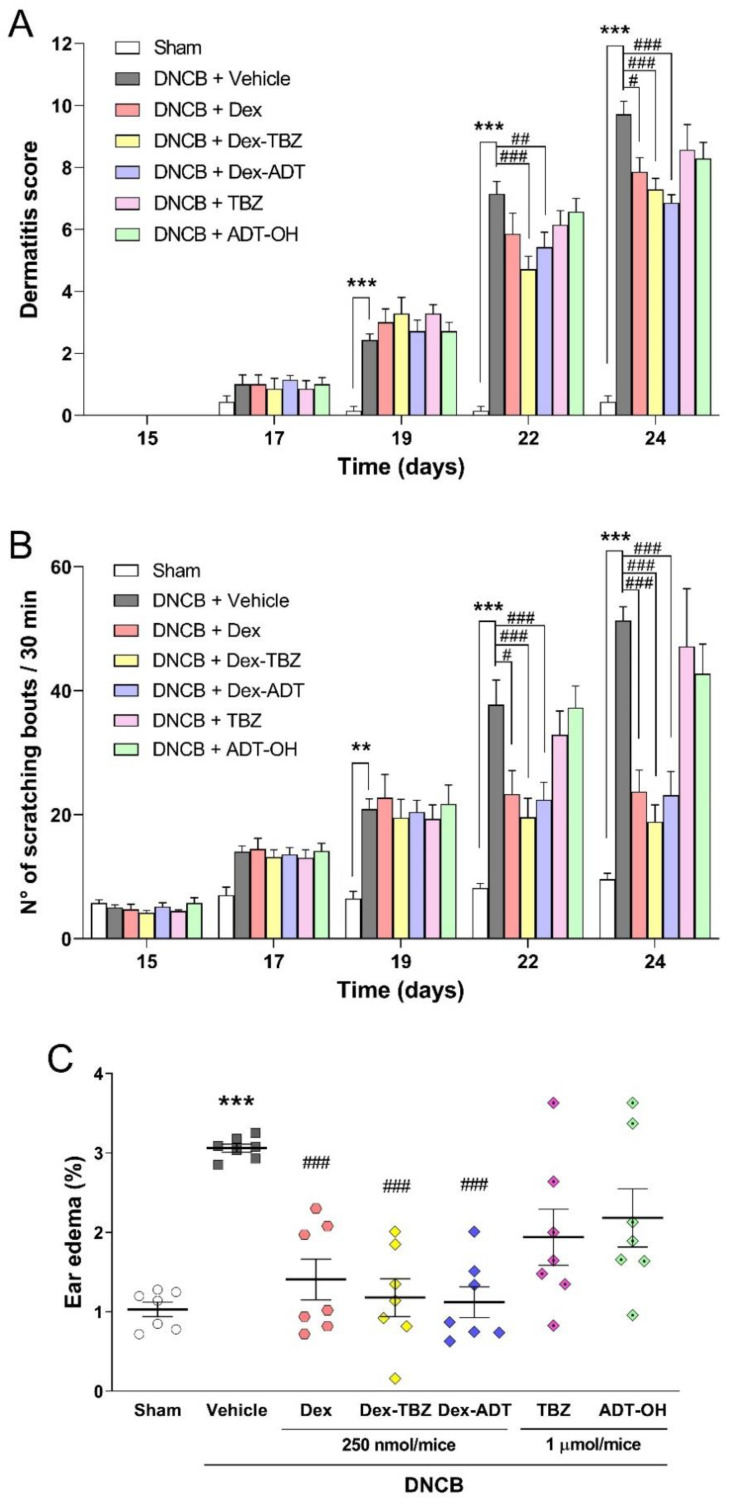
Effects of dexamethasone, H_2_S-releasing dexamethasone derivatives (Dex-TBZ and Dex-ADT), and H_2_S-releasing moieties (TBZ and ADT-OH) on DNCB-induced AD-like skin lesions (panel **A**; *n* = 7), scratching behavior (panel **B**; *n* = 7), and ear edema (panel **C**; *n* = 7). Data are expressed as mean ± SEM. ** *p* < 0.01 and *** *p* < 0.001 vs. Sham; ^#^
*p* < 0.05, ^##^
*p* < 0.01 and ^###^
*p* < 0.001 vs. DNCB + Vehicle.

**Figure 4 pharmaceutics-15-01907-f004:**
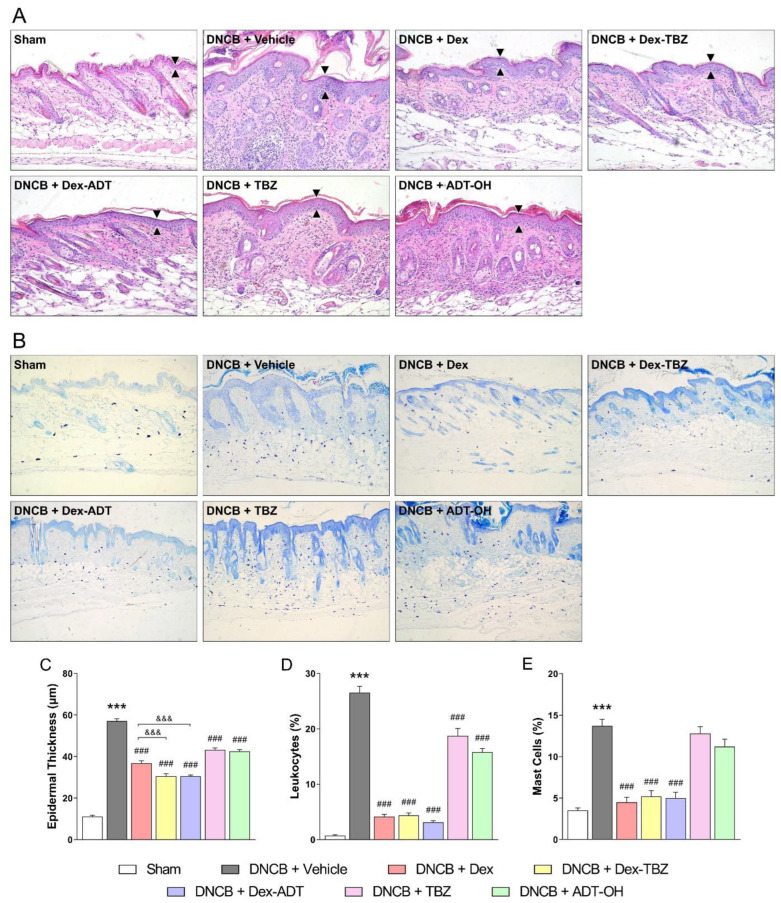
Effects of dexamethasone, Dex-TBZ, Dex-ADT, TBZ, and ADT-OH on the cutaneous histopathological findings in DNCB-induced atopic dermatitis in mice. Representative histological images of sections of cutaneous tissue stained with H&E and analyzed for epidermal thickness (black arrowhead) and leukocyte infiltration (200× magnification; panel **A**), sections of skin tissue stained with toluidine blue and analyzed for mast cell infiltration (200× magnification; panel **B**). Panel **C**: epidermal thickness (*n* = 5); panel **D**: leukocyte infiltration (*n* = 5); panel **E**: mast cell infiltration (*n* = 5). Data are expressed as mean ± SEM. *** *p* < 0.001 vs. Sham; ^###^
*p* < 0.001 vs. DNCB + Vehicle; ^&&&^
*p* < 0.001 vs. DNCB + Dex.

**Figure 5 pharmaceutics-15-01907-f005:**
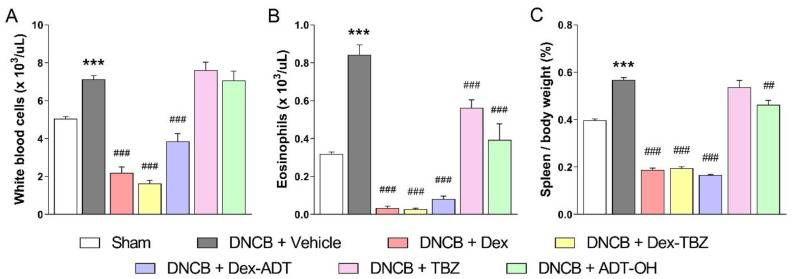
Effects of dexamethasone, Dex-TBZ, Dex-ADT, TBZ, and ADT-OH on DNCB-induced leukocytosis (panel **A**; *n* = 7), eosinophilia (panel **B**; *n* = 7), and splenomegaly (panel **C**; *n* = 7). Data are expressed as mean ± SEM. *** *p* < 0.001 vs. Sham; ^##^
*p* < 0.01 and ^###^
*p* < 0.001 vs. DNCB + Vehicle.

**Figure 6 pharmaceutics-15-01907-f006:**
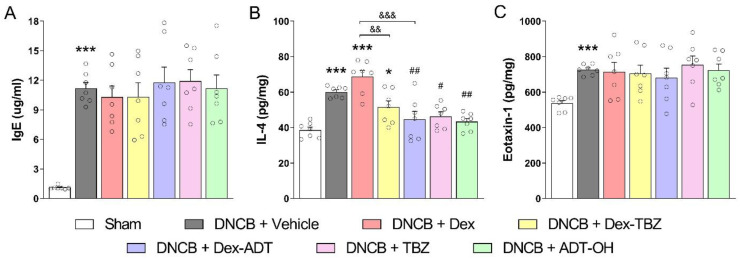
Effects of dexamethasone, Dex-TBZ, Dex-ADT, TBZ, and ADT-OH on serum IgE (panel **A**; *n* = 7) and skin IL-4 (panel **B**; *n* = 7) and eotaxin-1 (panel **C**; *n* = 7) contents. Data are expressed as mean ± SEM. * *p* < 0.05 and *** *p* < 0.001 vs. Sham; ^#^
*p* < 0.05 and ^##^
*p* < 0.01 vs. DNCB + Vehicle; ^&&^
*p*< 0.01 and ^&&&^
*p* < 0.001 vs. DNCB + Dex.

**Figure 7 pharmaceutics-15-01907-f007:**
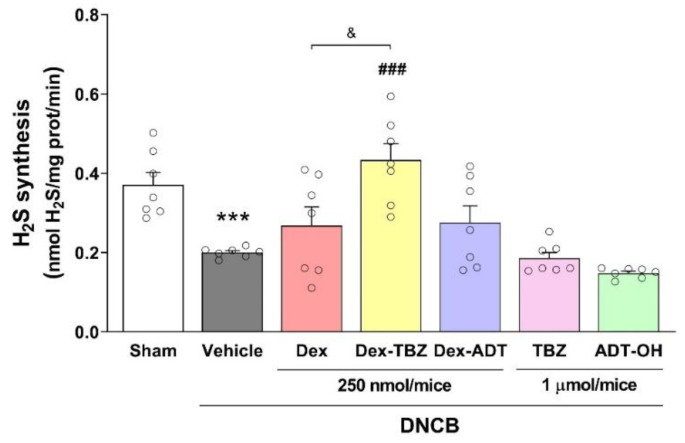
Effects of dexamethasone, Dex-TBZ, Dex-ADT, TBZ, and ADT-OH on endogenous H_2_S production in mouse skin. Data are expressed as mean ± SEM (*n* = 7). *** *p* < 0.001 vs. Sham; ^###^
*p* < 0.001 vs. DNCB + Vehicle; ^&^
*p* < 0.05 vs. DNCB + Dex.

**Figure 8 pharmaceutics-15-01907-f008:**
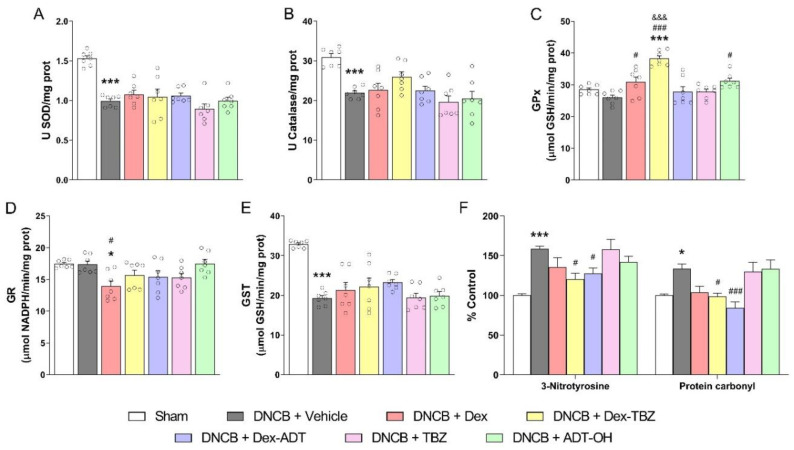
Effects of dexamethasone, Dex-TBZ, Dex-ADT, TBZ, and ADT-OH on the activity of the antioxidant enzymes SOD (panel **A**; *n* = 7), catalase (panel **B**; *n* = 7), GPx (panel **C**; *n* = 7), GR (panel **D**; *n* = 7), and GST (panel **E**; *n* = 7), and on the protein oxidative stress markers 3-nitrotyrosine and protein carbonylation (panel **F**; *n* = 7). Data are expressed as mean ± SEM. * *p* < 0.05 and *** *p* < 0.001 vs. Sham; ^#^
*p* < 0.05 and ^###^
*p* < 0.001 vs. DNCB + Vehicle; ^&&&^
*p* < 0.001 vs. DNCB + Dex.

**Figure 9 pharmaceutics-15-01907-f009:**
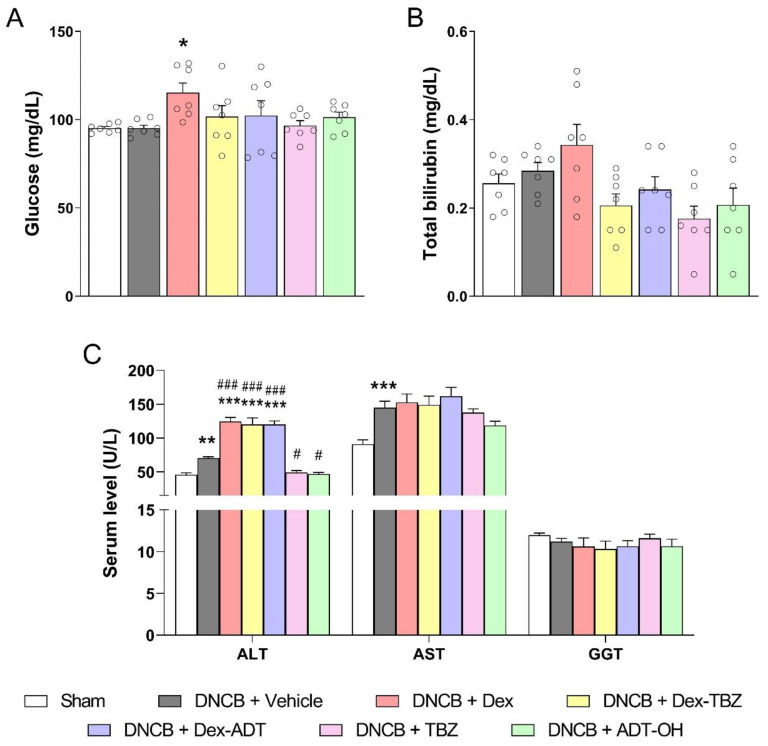
Effects of dexamethasone, Dex-TBZ, Dex-ADT, TBZ, and ADT-OH on the plasma concentrations of glucose (panel **A**; *n* = 7), bilirubin (panel **B**; *n* = 7), and liver enzymes (panel **C**; *n* = 7) of mice with DNCB-induced AD. Data are expressed as mean ± SEM. * *p* < 0.05, ** *p* < 0.01, and *** *p* < 0.001 vs. Sham; ^#^
*p* < 0.05 and ^###^
*p* < 0.001 vs. DNCB + Vehicle.

## Data Availability

The data presented in this study are available on request from the corresponding author.

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
