# Peer review of "Beneficial Effects of Two Hydrogen Sulfide (H2S)-Releasing Derivatives of Dexamethasone with Antioxidant Activity on Atopic Dermatitis in Mice"

_pharmaceutics, 2023, doi:10.3390/pharmaceutics15071907_

Round 1

Reviewer 1 Report

Very well written manuscript describing compiling evidences that H2S derivatives of dexamethasone can have beneficial effects on inflammatory skin diseases such as Atopic Dermatitis. No flaws were detected in the results presented and proper data interpretation was reported. 

Reviewer 2 Report

This manuscript deals with the use of H2S releasing dexamethasone derivatives aiming for atopic dermatitis (eczema) treatment. The manuscript is well written and the experiments are well conducted.

I recommend accepting the manuscript after performing the following:

1- In the abstract: Please change this statement: We conclude that the presence of H2S-releasing moieties in the Dex structure, not only does not interfere with the beneficial effects of this corticosteroid .....

2- Please provide the standard calibration curve of NaHS that was used in the calculation of H2S released.

3- Please determine the nature of the spheres present in figures 6, 7 and 8.

4- Please consider adding figures demonstrating the chemical structure of the H2S dexamethasone derivatives.

Just needs minor revision.

Reviewer 3 Report

The study evaluate the effectiveness of two H2S releasing derivatives of dexamethasone in preclinical model of atopic dermatitis. 

The results are very interesting and clearly presented. Methodology is proper and described in details. 

However, there are some minor issues that need clarification:

1. "Corticosteroids, such as dexamethasone, are considered the first-line therapy for control of allergic diseases, such as AD..."  Please refer to current AD treatment guidelines, e.g. EuroGuiDerm. Systemic corticosteroids are only used as rescue therapy and for a short time. This should be clearly marked. The first-line method are topical corticosteroids.

2. There are some points that need to be raised in the Discussion, such as:

- concentration of H2S in inflammatory skin diseases - are there data for atopic dermatitis and psoriasis?

- differences in H2S release depending on the donor (e.g. differences between Na2S and GYY);

- large differences in H2S concentration for different methods;

- pro- and antiinflammatory effects of H2S depending on the concentration.
